# Efficient Activation of Peroxymonosulfate by Cobalt Supported Used Resin Based Carbon Ball Catalyst for the Degradation of Ibuprofen

**DOI:** 10.3390/ma15145003

**Published:** 2022-07-18

**Authors:** Guangzhen Zhou, Yanhua Xu, Xiao Zhang, Yongjun Sun, Cheng Wang, Peng Yu

**Affiliations:** 1School of Environmental Science and Engineering, Nanjing Tech University, Nanjing 211816, China; guangzhenmail@126.com (G.Z.); yanhuaxu18@hotmail.com (Y.X.); zx383052679@163.com (X.Z.); yspong@163.com (P.Y.); 2College of Urban Construction, Nanjing Tech University, Nanjing 211816, China; sunyongjun@njtech.edu.cn

**Keywords:** used D001 resin, cobalt sulfide, carbon materials, PMS activation, ibuprofen

## Abstract

The extensive use of ibuprofen (IBU) and other pharmaceuticals and personal care products (PPCPs) causes them widely to exist in nature and be frequently detected in water bodies. Advanced catalytic oxidation processes (AOPs) are often used as an efficient way to degrade them, and the research on heterogeneous catalysts has become a hot spot in the field of AOPs. Among transitional metal-based catalysts, metal cobalt has been proved to be an effective element in activating peroxymonosulfate (PMS) to produce strong oxidizing components. In this study, the used D001 resin served as the matrix material and through simple impregnation and calcination, cobalt was successfully fixed on the carbon ball in the form of cobalt sulfide. When the catalyst was used to activate persulfate to degrade IBU, it was found that under certain reaction conditions, the degradation rate in one hour could exceed 70%, which was far higher than that of PMS and resin carbon balls alone. Here, we discussed the effects of catalyst loading, PMS concentration, pH value and temperature on IBU degradation. Through quenching experiments, it was found that ${\mathrm{SO}}_{4}{}^{-}$ and ·OH played a major role in the degradation process. The material has the advantages of simple preparation, low cost and convenient recovery, as well as realizing the purpose of reuse and degrading organic pollutants efficiently.

## 1. Introduction

A large number of so-called pharmaceutical and personal care product pollutants (PPCPs) have been a class of persistent organic pollutants around the world due to people’s extensive use and its difficult degradation in the water environment, which has become a concern of the global community [1]. Ibuprofen (IBU) as a typical PPCP, used to treat pain, fever as an anti-inflammatory with the advantages of low toxicity, fine curative effect and few side effects, has become a new generation of non-steroidal anti-inflammatory drugs and is widely used [2,3]. Massive abuse and continuous input of PPCPs make them a false persistence phenomenon in the environment. The long-term ingestion of trace levels of these substances by organisms caused biological malformations and microbial drug resistance, indirectly harming the aquatic ecological environment and human health and receiving extensive attention [4]. However, traditional water treatment processes, such as sedimentation, filtration and disinfection have poor effects on the separation of IBU [5,6,7]. At present, the commonly used treatment methods include the adsorption method [8], membrane separation method [9], biological method [10], photocatalysis (Visible or ultraviolet catalysis) [11,12,13], Fenton Catalytic oxidation and ozone catalytic oxidation [14,15], etc., advanced oxidation methods (AOPs).

Advanced oxidation catalysts, especially heterogeneous catalyst materials used to catalyze the formation of free radicals, have been widely studied in wastewater treatment. Compared with Fenton, Fenton-like and ozone catalytic oxidation technologies, the sulfate radical (${\mathrm{SO}}_{4}{}^{-}\cdot$) produced by activating peroxymonosulfate (PMS) has higher redox potential [16,17], and PMS is stable and convenient for transportation and storage, which shows great application potential in the treatment of refractory organic pollutants [18]. Research shows that the production of ${\mathrm{SO}}_{4}{}^{-}\cdot$ activated PMS plays a major role, at the same time, the function of the hydroxyl radical (·OH), superoxide radical ($O_{2}{}^{-}$) and singlet oxygen (O12) cannot be ignored [4,7]. Therefore, increasing the rate of PMS decomposition to produce free or nonfree radicals has become a research hotspot. Currently available activation methods of PMS include thermal activation, electrical activation [19], photoactivation and heterogeneous catalyst material activation [4,20]. Although heterogeneous catalysts have been widely studied [21,22], if the materials are still in a powder state, especially without magnetism [17,23], solid–liquid separation and recovery would still be difficult. Therefore, the preparation of a certain shape catalyst with practical application value is significant, especially active components supported by tangible matrix materials, such as resin [24], alumina [25], sponges [26], etc.

Among various metal-based catalytic materials used to activate PMS [22], the standard electrode potential of ${\mathrm{Co}}^{2+}/{\mathrm{Co}}^{3+}$ (1.82 V) is higher than other metals [27] and close to that of PMS (1.82 V) [17] and it has the splendid quality to activate PMS and has attracted increasing attention. Its corresponding metal oxides [23], hydroxides [6] and sulfides [28,29] are widely used in the decomposition of activated persulfate; cobalt is the most widely studied metal element doped in heterogeneous materials at present. Moreover, studies have shown that sulfur-containing catalysts have many positive effects in the catalytic reaction process [30,31] with the help of accelerating electron transfer, which results from an abundance of electrochemically active sites available for the adsorption or desorption of $O^{2-}/H^{+}$ as well as their favorable electrical configuration [1,32]. This may also be the reason that transition metal sulfide proved to be used a relatively novel catalytic material to effectively catalyze and activate PMS to degrade organic pollutants in water [28,33]. 

D001 resin is a cation exchange resin with a sulfonic acid group ($-{\mathrm{SO}}_{3}H$) on a styrene divinylbenzene copolymer with a porous structure. At present, it is commonly used in pure water softening to adsorb calcium, magnesium ions, etc., and impurities in water. When the D001 resin was repeatedly used, its performance could not meet the requirements of use. Then, it often would be abandoned and not used further. However, this used resin still had some adsorption capacity. Through experiments, it could be found that used resins can adsorb cobalt ions and exchange metal ions, such as calcium and magnesium. The content of carbon, sulfur and oxygen accounts for the majority of D001 resin in the previous elemental analysis, and the corresponding metal oxide or sulfide can be formed by calcination with a transition metal. This means that the metal adsorbed on the resin through cation exchange and finally sintered at high temperature may exist on the carbon material in the form of sulfide or oxide, which indirectly synthesizes the metal-based carbon material.

Based on the above research and description, we used D001 soft water resin which adsorbed certain metal ions, such as calcium and magnesium ions and is regarded as a metal anchored carrier. It can adsorb cobalt ions on the resin through simple impregnation and is sintered and fixed on the resin through a tubular furnace under the protection of nitrogen. A series of characterizations were carried out to recognize the carbon material, and its external morphology, material composition, element valence, and specific surface area were analyzed. Finally, the catalyst activity was verified by the experiment of activating PMS to degrade IBU. Here, we explored the effects of catalyst loading, PMS concentration, pH value, temperature and four kinds of coexisting anions on the degradation effect of IBU separately. The possible mechanism and path in the PMS activating system under the action of a carbon ball catalyst were analyzed by free radical and nonfree quenching and related characterization.

## 2. Materials and Methods

### 2.1. Materials and Instruments

(1) Main drugs and reagents: Used D001 resin (Jinkai resin company, Shanghai, China), Cobalt nitrate hexahydrate (CAS: 10026-22-9, AR, Sinopharm chemical reagent, Shanghai, China), Potassium peroxymonosulfate (CAS: 70693-62-8, AR, Sinopharm chemical reagent), IBU (CAS: 15687-27-1, AR, Shanghai aladdln biochemical technology, Shanghai, China), Methanol (Meth, CAS: 67-56-1, AR, Sinopharm chemical reagent), Ethanol (Eth, CAS: 64-17-5, AR, Sinopharm chemical reagent), Tert butyl alcohol (TBA, CAS: 75-65-0, AR, Sinopharm chemical reagent), Furfuryl alcohol (FFA, CAS: 98-00-0, China Huixing biochemical reagent, Shanghai, China), P-benzoquinone(p-BQ, CAS: 106-51-4, 99%, Shanghai aladdln biochemical technology, Shanghai, China), Sodium bicarbonate (CAS: 144-55-8, GR, Nanjing Chemical Reagent, Nanjing, China), Anhydrous sodium sulfate (CAS: 7757-82-6, AR, Sinopharm chemical reagent), Sodium dihydrogen phosphate (CAS: 13472-35-0, AR, Shanghai Lingfeng chemical reagent, Shanghai, China), Sodium chloride (CAS: 7647-14-5, AR, Xilong scientific and chemical experimental reagent, Shanghai, China), etc.

(2) Main instruments: Water bath thermostatic oscillator (SHA-B and SHA-C, GuoHua enterprise, Changzhou, China), Thermostatic drying oven (DHG-9030, Shanghai Yiheng drying oven, Shanghai, China), Tubular furnace (TL-1200, Nanjing Boyuntong tubular furnace, Nanjing, China), pH meter (PHS-3C, Shanghai Leici, Shanghai, China), High performance liquid chromatograph (HPLC, LC-20AT, Shimadzu, Tokyo, Japan), Total organic carbon analyzer (TOC-LCPH, v-100~240, Shimadzu, Tokyo, Japan), Inductively coupled plasma optical emission spectroscopy (ICP-OES, Agilent 5100, New York, USA), Scanning electron microscope (SEM, Zeiss Gemini 300, Berlin, Germany), Electron emission spectrometer (EDS, Smart EDX and super-x, Berlin, Germany), Transmission scanning electron microscope (TEM, Fei Talos F200x G2, New York, USA), X-ray diffractometer (XRD, max-TTR-III Rigaku, Tokyo, Japan), Fourier transform infrared (FT-IR, IRAffinity-1S, Shimadzu Corporation, Japan). X-ray photoelectron spectrometer (XPS, Thermo Scientific k-alpha, New York, NY, USA), Brunauer Emmett Teller (BET, Tristar-3020, New York, NY, USA), etc.

### 2.2. Material Preparation

(1) Put the used D001 resin into a conical flask and add an appropriate amount of ethanol, then put it into a constant temperature oscillator to oscillate for several hours. Pour out the solution, add an appropriate amount of water and oscillate; repeat this washing two to three times until the washing water is clear, take out the resin and dry it.

(2) Impregnate a certain molar amount (0.25–1.25 mmol) of cobalt nitrate hexahydrate solution per gram of used resin, with a solid–liquid ratio of 1 g:20 mL. After shaking for two hours, take out the resin and dry it to obtain the used resin adsorbed with cobalt ions.

(3) The catalyst was prepared by calcining cobalt doped resin in a tubular furnace at 550 °C under the protection of nitrogen for 6 h, with a heating rate of 5 °C/min. The black sphere formed by sintering is the catalyst material we need.

### 2.3. Analysis Method

The surface morphology and microstructure of the material were studied by scanning electron microscope (SEM) and a transmission electron microscope (TEM) was applied for morphology and crystal structure identification. It was equipped with an energy dispersive X-ray detector (EDS) to obtain the different elements mapping graph. Then the possible material composition and its crystal surface or crystal form were characterized and analyzed by means of X-ray diffraction (XRD) with a Cu-Kα radiation source working at 36 kV and 20 mA and Fourier transform infrared spectroscopy (FT-IR) based on sample preparation with dry potassium bromide to detect the possible functional groups and chemical bonds on carbon ball materials. The valence state of elements and possible chemical bond compositions were calculated by fitting after the valence state of the elements is characterized by X-ray photoelectron spectroscopy (XPS) equipped with a dual X-ray source of Al-Kα (hv = 1486.6 eV). Finally, the specific surface area and pore diameter were analyzed by Brunauer Emmett Teller (BET) based on nitrogen adsorption–desorption isotherm measurements at −196 °C.

The catalytic activity of the sintered carbon ball was verified by degrading 50 mL IBU solution with a concentration of 10 mg/L in an oscillating conical flask. We aspirated a 2 mL water sample with syringes and filtered it through a 0.22 µm filter head (a few drops were filtered out to remove the residual liquid from the filter head of the previous sample and then injected into a 1.5 mL liquid phase sample bottle containing 10 µL methanol to stop the subsequent reaction) every 10 min and detected by HPLC. The specific detection conditions were as follows: the mobile phase is acetonitrile (70%) and Wahaha water (30%, adjusted pH = 2 with phosphoric acid), the chromatographic column used was a C18 reverse column (5 µm, 4.6 mm × 250 mm, Shimadzu); the detected temperature was 35 °C, we chose 1 mL/min as the liquid flow rate, and the injection volume of every sample was 20 µL. Then we used 221 nm as the analysis wavelength to obtain the corresponding peak area. Through the previous liquid phase determination of the standard solution, it can be found that the peak area tested by this method had a high degree of linear correlation with the concentration, so the degradation rate of IBU could be calculated according to the measured peak area. Based on the above methods, we studied the catalytic degradation of IBU in different systems and different cobalt doping. At the same time, the effects of several environmental conditions, such as catalyst loading, PMS concentration, temperature, pH value, and coexisting anions, on the catalytic activity and IBU degradation rate were similarly studied. Finally, the corresponding quenchers were used to capture possible free radicals and non-free radicals (MeOH was used to capture both ${\mathrm{SO}}_{4}{}^{-}$· and OH, TBA, p-BQ and FFA were applied to quench OH, $O_{2}{}^{-}$ and O12, respectively. We did not choose L-histidine as the quencher of singlet oxygen because it may react with PMS) [23,31,34]. Combined with relevant characterization and previous studies, the causes and degradation mechanism were reasonably analyzed and some strange experimental phenomena can also be explained.

## 3. Results and Discussion

### 3.1. Material Characterization

First, we characterized the surface morphology of the material. The material morphology photograph under high magnification of SEM is shown in Figure 1. Meanwhile, we scanned the shooting site for relevant elements, and the EDS results are shown in Appendix A. We find that the element of sulfur and carbon occupied most of the carbon ball, and the distribution of cobalt can also be clearly seen in the carbon ball doped with abundant cobalt, both inside and outside. According to SEM results, Figure 1a shows that the material presents a sphere after high-temperature carbonization, but there also exists a broken globule after the calcination process. Figure 1b shows that the surface of the noncarbonized used D001 resin has a porous morphology, which is conducive to its efficient adsorption of metal ions as a cation exchange resin. However, we can see from Figure 1c that after carbonization of the used resin, a large number of blocks and strips are attached to the surface of the resin, which means that calcium and magnesium form chemical compounds on the surface. In contrast, Figure 1d shows an obvious change in the surface of the carbon ball, which is covered by a large number of small particles and even blocks many pores on the resin surface. Compared with the former, there are no long strips and blocks, thus a large number of calcium and magnesium ions are replaced by cobalt ions, and the surface is covered by metal composites sintered on it. Figure 1e,f show the internal morphology of spheres after cobalt doping calcination. It can be seen from Figure 1f, that although many surface pores of the resin are covered after high-temperature calcination, the interior still shows a porous morphology. Compared with Figure 1d,f, it is found that metal cobalt is successfully attached to the surface of the resin and the interior of the resin pores. Combined with EDS analysis, we found that there are mainly sulfur and oxygen elements in the carbon ball, so the burned metal substances are the corresponding metal sulfides or oxides.

As shown in Figure 2, further investigation of the carbonized catalyst material was carried out by TEM images at higher magnification. It can be seen from the morphology in Figure 2a that there are multiple blocks on the carbon material, which may be the compound of cobalt sintered on the carbon material. The mapping diagrams of elemental sulfur and elemental cobalt under the corresponding morphology in Figure 2b,c clearly show that the delamination positions of these two elements are similar, which can prove that after the used resin adsorbs cobalt ions, the metal cobalt can be sintered with the $-{\mathrm{SO}}_{3}H$ in the resin to form a metal sulfide. Similarly, the same conclusion can be obtained by combining the morphology of Figure 2d and the element layered images of Figure 2e,f. Therefore, it can be inferred that SEM is attached to the carbon ball. 

In Figure 3, it can be seen from the XRD diagram of used resin and high-temperature burning after impregnation with different concentrations of cobalt ions, that all the carbon ball materials have typical carbon peaks at 2θ values of around 20° [35]. After carbonization of the used resin alone, diffraction peaks appear at diffraction angles of 42.9° and 62.4°, respectively, which may be magnesium oxide formed by calcination, corresponding to (200) and (220) crystal planes of magnesium oxide standard card JCPDS: 45-0946. For the material doped with metal cobalt, the diffraction peaks at these two positions are greatly weakened or even disappear, indicating that the magnesium ions in the resin may be replaced by cobalt ions in the process of impregnating cobalt ions. The characteristic diffraction peaks of cobalt sulfide appear at the diffraction angles of 30.5°, 35.2°, 47.7° and 54.5°, respectively, which correspond to the (100), (101), (102) and (110) crystal planes [28,36] of the standard card JCPDS: 48-0826, respectively. According to the variation trend of peak type and cobalt concentration and the position of diffraction peak, we can infer that metal cobalt sintered on the surface, or inside of carbon ball, in the form of cobalt sulfide.

To determine the approximate estimation of the element valence and content of the prepared carbon spherical catalyst, X-ray photoelectron spectroscopy (XPS) was performed. In the early stage, we carried out the characterization of FT-IR to determine the possible chemical bonds or functional groups in the material. The results are shown in Appendix A. It can be found that after cobalt doping, the obvious peak positions of the material did not change, that is, they all have characteristic absorption in similar positions, stretching vibration peaks appear at 3433, 1580, 1400, and 1130 cm^−1^ which can be associated with O-H, carbonyl (C-C/C=C), C-S and C-O, respectively [31,37]. Based on this, we obtain Figure 4a–d corresponding to the valence binding energy peaks of several main elements XPS data graph after our date processing on behalf of sulfur (S 2p), carbon (C 1s), oxygen (O 1s) and cobalt (Co 2p), respectively. In Figure 4a, the characteristic peaks of C-S or C-${\mathrm{SO}}_{x}$ appeared in the binding energy of 164 eV, 165 eV and 167.8 eV [31], while there are characteristic peaks of spin orbits corresponding to S 2$p_{3/2}$ and S 2$p_{1/2}$ at 161.9 eV and 163.2 eV, respectively [36,38,39]. The position and area of each binding energy peak of C 1s show that it mainly exists in the form of C-C/C=C [37,40,41], and at 285.3 eV may represent C-S binding energy [42]. At other binding energy positions, it may correspond to a small amount of C-O or C=O [37,41,43]. Figure 4c shows the peak splitting results of the elemental oxygen spectrum, it can be found that the peak area of the oxygen element is mainly at the position with a binding energy of 532.1 eV, which indicates that oxygen is mainly a carbon–oxygen bond [43] or surface adsorbed oxygen species ($O_{\mathrm{aos}}$) [41]. Figure 4d shows the diffraction peaks of several different binding energy positions of cobalt. The binding energy positions at 785.1 eV and 802 eV may be two shake-up satellite peaks generated by X-ray emission during sample measurement [44,45,46]; the binding energy positions at 778.5 eV and 780.8 eV, are Co 2$p_{3/2}$ spin orbit trivalent cobalt and divalent cobalt, respectively; 793.6 eV and 797.0 eV correspond to trivalent cobalt and divalent cobalt in Co 2$p_{1/2}$ spin-orbit peaks [44,47]. Compared with the peak area, we can find that the peak area of divalent cobalt is much larger than that of trivalent cobalt, it means that the main valence states of cobalt are divalent. From what has been discussed about the XPS peak splitting results of sulfur, we can conclude that the cobalt ions adsorbed on the resin are mainly sintered on the carbon ball in the form of cobalt sulfide.

Finally, we analyzed and tested the parameters related to the specific surface area of the material, the nitrogen adsorption−desorption curve is shown in Appendix A. As shown in Table 1, we listed the specific surface area, pore volume, pore diameter and other data of D001 resin, used D001 resin and after cobalt doped carbonized material. The measured mass of the samples is 0.1 g and the degassing time of pretreatment was 6 h, and then the sample carried out nitrogen adsorption and desorption in a liquid nitrogen tank under −196 °C. Finally, the relevant characterization data were calculated according to the nitrogen adsorption>–desorption curve. It can be found that the specific surface area of the three samples shows a downward trend with the doping of metal, and the values are 97.62 m^2^/g, 67.44 m^2^/g and 31.54 m^2^/g, respectively. In particular, we have tested the specific surface area of the maximum cobalt doping amount (impregnated with 1.25 mmol cobalt ion per gram of resin), and the value is less than 20 m^2^/g; Appendix A also shows that the adsorption capacity of sintered carbon ball decreases with the increase in cobalt loading. According to the scanning electron microscope characterization and relevant experience, this phenomenon can be speculated and explained as that there is no metal doped in the separate resin. After calcination, the resin sphere is heated, and the skeleton shrinks, but the internal channels are not blocked. In contrast, the adsorbed water and impurities will separate from the channels due to calcination, so it has a large specific surface area. However, a large number of metal ions are adsorbed inside the resin, then metal complexes may be formed inside the resin during calcination. Especially after impregnating cobalt ions, internal cobalt ions are attached to the internal pores of the resin in the form of sulfide, and the surface pores are seriously blocked due to the existence of fine cobalt sulfide, which greatly reduces the specific surface area of the carbon ball. Moreover, the active sites of the internal metal cobalt are more difficult to expose with more cobalt content, resulting in the decline of its catalytic effect.

### 3.2. Material Catalytic Activity

Firstly, we explore the degradation effect of IBU under different systems. Figure 5a shows the catalytic performance of carbon ball catalysts prepared with different cobalt loading. It can be seen that the degradation effect of separate resin carbon ball and PMS system on IBU is poor, which also proves that the degradation or adsorption performance of separate PMS and resin system for IBU are similarly low. In previous studies, we found that under the same conditions, the material prepared by doping other transition metals into the resin has a very poor effect on activating PMS to degrade IBU, while the cobalt doped has an efficient catalytic effect, and its degradation effect can exceed 70% in one hour. The degradation process followed the quasi-first-order kinetic model after kinetic analysis, and its correlation coefficient of ln (C_0_/C) and time are basically above 95%, and the maximum kinetic constant (k value) is 0.216 min^−1^. Combined with the catalytic effect of catalysts prepared with different cobalt doping amounts, and the residual rate of cobalt ions after resin adsorption and saturation in the preparation process, as shown in Figure 5b, the optimal doping amount is 0.75 mmol cobalt ions doped per gram of dry resin; the residue rate at this impregnation amount is less than 5% which is analyzed by ICP-OES. In terms of activity, increasing the doping amount of cobalt may reduce the activity by blocking more internal pores, which is not conducive to the exposure of the internal cobalt. According to the impregnation results, cobalt ions are basically impregnated, and the mass of cobalt ions remains unchanged after calcination, while the mass loss of resin into carbon spheres is about half, so the mass fraction of cobalt can be calculated with the amount of impregnated cobalt and the mass of carbon spheres after calcination; the mass fraction of metal cobalt is estimated to be 9% in optimal doping amount, and the subsequent activity and mechanism are studied with this doping amount.

In terms of reaction conditions, we discussed the effects of catalyst loading, PMS concentration, pH value and temperature on the degradation effect, respectively, and the results are shown in Figure 6. It can be seen that under the same conditions as other variables, with the increase in catalyst loading, PMS concentration and temperature can improve the degradation effect of IBU. Comprehensively considering the catalytic degradation effect and reaction conditions, we regarded the catalyst loading of 0.3 g/L, PMS concentration of 0.7 mmol/L as the optimum catalyst and oxidant conditions and a temperature of 25 °C to fit the ambient temperature to study the effects of various influencing factors on the activation of PMS by a catalyst. Among all studied factors, the degradation process is greatly affected by pH value. When pH is 5 and 9, the degradation rate is decreased, while when pH is equal to 3 and 11, the degradation rate of IBU is significantly inhibited, and its kinetic constant is as low as 0.085 min^−1^ and 0.099 min^−1^. According to relevant studies, the reason for this phenomenon may be that under the conditions of low pH values, anions (${\mathrm{Cl}}^{-}$) have a certain inhibition [48], more importantly, acidic conditions are not conducive to the ionization of PMS [6,28], and a relatively high concentration of $H^{+}$ may quench a portion of the radicals [23]. Under the condition of a high pH value, relevant research shows that it will inhibit the reaction direction of cobalt sulfide and hydrogen persulfate ions to form ${\mathrm{SO}}_{4}{}^{-}$, due to the formation of ${\mathrm{OH}}^{-}$ in the reaction process; even the ${\mathrm{SO}}_{4}{}^{-}$ generated from PMS can react with ${\mathrm{OH}}^{-}$ and create ·OH with low activity, as shown in the reaction Formulas (4) and (11) [30,48].

Next, we also studied the influence of different concentrations of coexisting anions on the environmental conditions and discussed a certain concentration of ${\mathrm{HCO}}_{3}{}^{-}$, $H_{2}{\mathrm{PO}}_{4}{}^{-},\;\;{\mathrm{SO}}_{4}{}^{2-},\;{\;\mathrm{Cl}}^{-}$ in the reaction system, as shown in Figure 7. It can be seen that the low concentration of bicarbonate ion has no obvious inhibition on the reaction, while the high concentration does differ. Through the determination of pH value and comparison with the previous pH value influence experiment, it can also be found that the inhibition effect of bicarbonate ion is not large, and mainly through increasing the pH value of water. Similarly, with the addition of $H_{2}{\mathrm{PO}}_{4}{}^{-}$, the pH value decreases, but the removal rate of IBU does not decrease much, which can better prove that $H_{2}{\mathrm{PO}}_{4}{}^{-}$ has no obvious inhibitory effect. ${\mathrm{SO}}_{4}{}^{2-}$ has a certain degree of inhibition on the reaction system. Combined with relevant studies, it can be explained that the existence of ${\mathrm{SO}}_{4}{}^{2-}$ will inhibit the formation of ${\mathrm{SO}}_{4}{}^{2-}$ by ${\mathrm{HSO}}_{5}{}^{-}$, which will reduce the formation rate of OH or $O_{2}{}^{-}$ indirectly as shown in the reaction Formulas (5) and (7) [3]. The influence of ${\mathrm{Cl}}^{-}$ is very obvious, the inhibition rate is as high as 60%, and the final effect is not much better than that of the resin and PMS degradation system, more importantly, the kinetic constants are all around 0.04 min^−1^, and the linear correlation coefficient of the first-order kinetic fitting is only 85%. This phenomenon is similar to the study on activating PMS with cobalt doped carbon matrix catalyst prepared by Ren Z.F. [48]. This may be because the PMS concentration in the reaction system is only 0.7mmol, while the concentration of ${\mathrm{Cl}}^{-}$ exceeds 1mmol, and IBU is an organic pollutant that is difficult to degrade. Under these conditions, ${\mathrm{SO}}_{4}{}^{2-}$ or ${\mathrm{HSO}}_{5}{}^{-}$ reacts with ${\mathrm{Cl}}^{-}$ to generate a chloride ion radical with low oxidation performance, and its ability to degrade pollutants is poor [31]. However, with the increased concentration of ${\mathrm{Cl}}^{-}$, the generation rate of chloride-related radicals accelerates, but its oxidation performance is low, and it may not show a degradation effect. The reaction may be shown in Formulas (1)–(3) [48].
(1)$${\mathrm{Cl}}^{-}+{\mathrm{SO}}_{4}{}^{-}\to \mathrm{Cl}\;+{\;\mathrm{SO}}_{4}{}^{2-},$$
(2)$${\mathrm{Cl}}^{-}+\mathrm{Cl}\to {\mathrm{Cl}}_{2}{}^{-},$$
(3)$${\mathrm{Cl}}_{2}{}^{-}+H_{2}O\to {\mathrm{ClOH}}^{-}+H^{+}+{\mathrm{Cl}}^{-},$$

### 3.3. Mechanism Exploration

In the part of mechanism exploration, we studied the catalyst activating PMS to degrade IBU through relevant free or nonfree radical quenching experiments as Figure 8 shown. From the quenching experiment, it can be seen that by adding a certain amount of methanol quencher to the reaction system, the degradation effect of IBU is basically reduced to be similar to that of PMS alone, and the kinetic constants decreased from 0.216 min^−1^ to 0.043 min^−1^ and 0.128 min^−1^ under the maximum quenching degree by MeOH and TBA, respectively. Comparing the quenching effect of $O_{2}{}^{-}$ and $O_{2}{}^{1}$, it can be concluded that the degradation process of IBU is mainly played by ${\mathrm{SO}}_{4}{}^{-}$ and OH [49], while the other two have no obvious effect on the reaction process, which proves that the cobalt sulfide existing on the carbon ball reacts with the ${\mathrm{HSO}}_{5}{}^{-}$ ionized by PMS to form abundant ${\mathrm{SO}}_{4}{}^{-}$ and OH; relevant possible reaction Formulas are 4 and 5 [17]. We also find that the effect of ${\mathrm{SO}}_{4}{}^{-}$ is much better than OH. it may be that the ${\mathrm{SO}}_{4}{}^{-}$ generated by the reaction accounts for the majority. On the other hand, its redox potential exceeds OH [16]. At the same time, it can prove that the generated amount of $O_{2}{}^{-}$ and O12 is small in this degradation system, in particular, O12 is based on generated by $O_{2}{}^{-}$. Based on this and combined with existing research, relevant possible reactions as Formulas (4)–(12) are shown [3,50,51]:
(4)$${\mathrm{Co}}^{2+}+{\mathrm{HSO}}_{5}{}^{-}\to {\mathrm{Co}}^{3+}+{\mathrm{SO}}_{4}{}^{-}+{\;\mathrm{OH}}^{-},$$
(5)$${\mathrm{Co}}^{2+}+{\;\mathrm{HSO}}_{5}{}^{-}\;\to {\;\mathrm{SO}}_{4}{}^{2-}+{\;\mathrm{Co}}^{3+}+\;\mathrm{OH},$$
(6)$${\mathrm{HSO}}_{5}{}^{-}\to {\mathrm{SO}}_{5}{}^{2-}+H^{+},$$
(7)$${\mathrm{SO}}_{5}{}^{2-}+H_{2}O\to O_{2}{}^{-}+{\mathrm{SO}}_{4}{}^{2-}+2H^{+},$$
(8)OH+O2−→O12+OH−,
(9)$${\mathrm{Co}}^{3+}+{\mathrm{HSO}}_{5}{}^{-}\to {\mathrm{Co}}^{2+}+{\mathrm{SO}}_{5}{}^{-}+H^{+},$$
(10)$$2{\mathrm{SO}}_{5}{}^{-}\to 2{\mathrm{SO}}_{4}{}^{-}+O_{2},$$
(11)$${\mathrm{SO}}_{4}{}^{-}+{\mathrm{OH}}^{-}\to {\mathrm{SO}}_{4}{}^{2-}+\cdot \mathrm{OH},$$
(12)IBU+SO4−/OH/→H+/O2−/O12→Oxidationproducts

The formation and degradation mechanism of free radicals is roughly shown in Figure 9 [17]. In general, after absorbing a certain amount of cobalt ions by impregnation, we carbonize the used resin into a black carbon ball. Through XRD and XPS, it can be concluded that cobalt is successfully attached to the carbon ball in the form of cobalt sulfide in the characterization of XRD and XPS. Through BET characterization, it can be found that the increase in cobalt may seriously block the carbon ball, which is not conducive to the exposure of internal cobalt sulfide, so the reaction speed is reduced. In the reaction system, the carbon ball activates PMS to produce a large amount of ${\mathrm{SO}}_{4}{}^{-}$ and OH degrade IBU into small molecular substances, ${\mathrm{CO}}_{2}$ and $H_{2}O$, but other active substances generated from this reaction are extremely few.

## 4. Conclusions

Utilizing the used D001 resin as the base material, the metal cobalt is fixed on the resin by impregnation, and then the cobalt is sintered on the carbon ball in the form of cobalt sulfide by high-temperature calcination. The preparation method is simple, low cost and has low material loss with a high resource degree. In terms of material activity, under optimal cobalt doping and certain reaction conditions, the degradation rate of IBU can exceed 70%. To a certain extent, the increase in catalyst loading, PMS concentration and temperature can improve the reaction rate of the degradation system. However, the material is greatly affected by the environmental conditions in the reaction process, especially the pH value and the existence of several common anions. According to the free and non-free radical quenching experiment, it is not difficult to see that the degradation process of IBU is mainly the function of ${\mathrm{SO}}_{4}{}^{-}$ and OH produced by catalyst-activated PMS.

## Figures and Tables

**Figure 1 materials-15-05003-f001:**
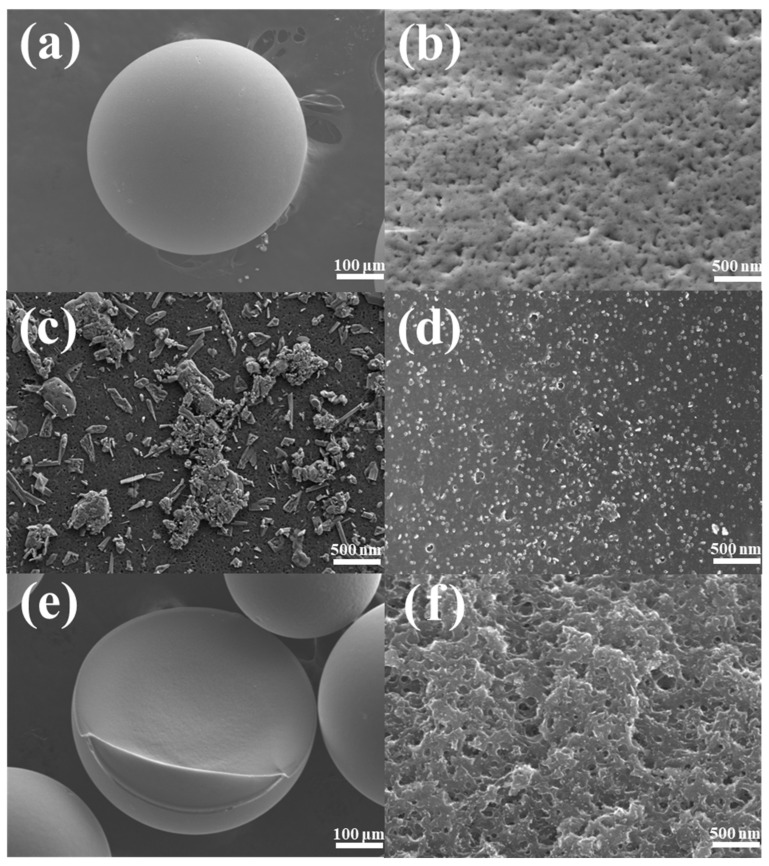
The surface morphology of a single resin or carbon ball (**a**), used resin (**b**), carbonized used resin (**c**) and carbonized cobalt doped used resin (**d**), The internal cross-sectional morphology of carbonized cobalt doped used resin (**e**,**f**).

**Figure 2 materials-15-05003-f002:**
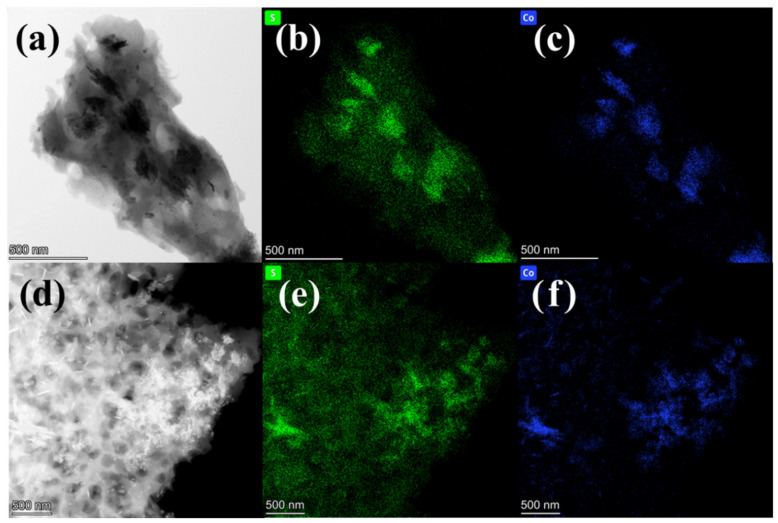
TEM diagrams of cobalt supported used resin based carbon ball catalyst (**a**,**d**), EDS layered mapping diagram of sulfur and cobalt elements (**b**,**c**,**e**,**f**).

**Figure 3 materials-15-05003-f003:**
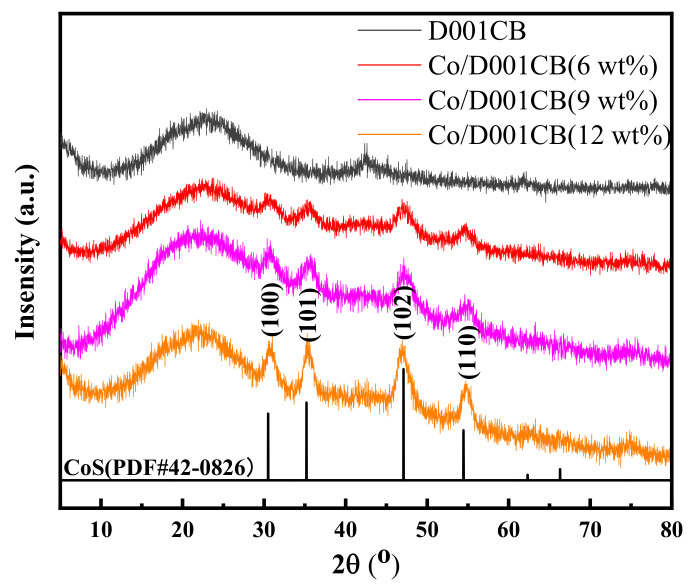
The XRD of carbon ball with different cobalt doping.

**Figure 4 materials-15-05003-f004:**
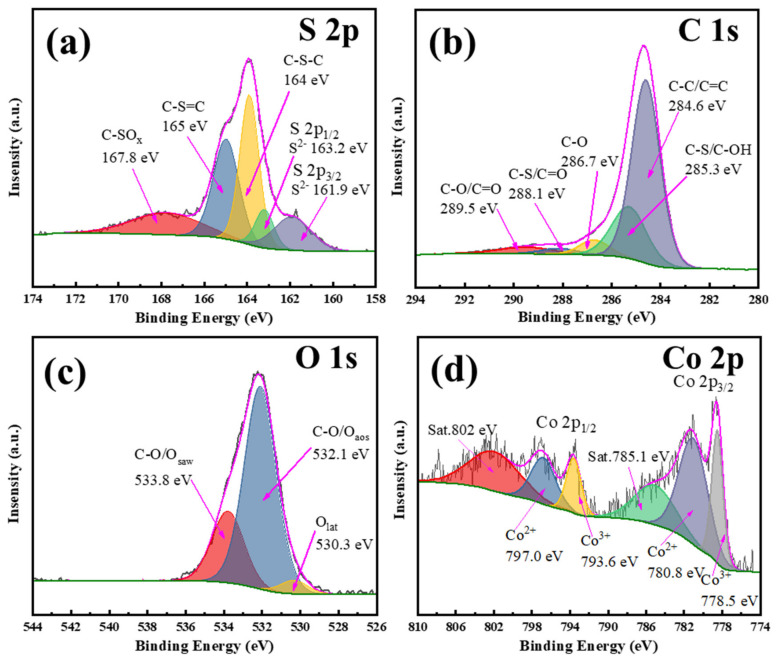
The XPS diagram of main elements of cobalt doped carbon sphere material, S 2p (**a**), C 1s (**b**), O 1s (**c**), Co 2p (**d**).

**Figure 5 materials-15-05003-f005:**
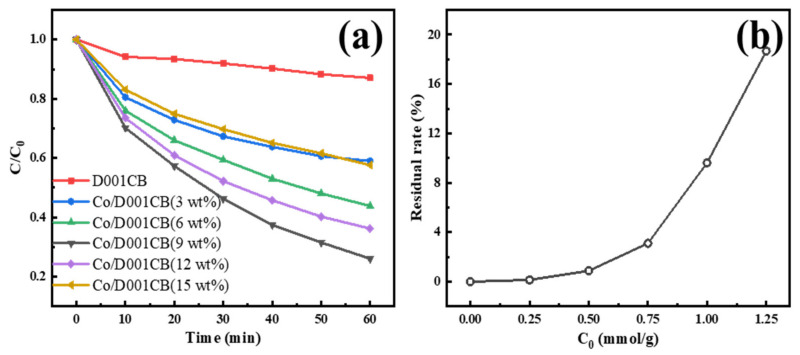
The curve of ibuprofen degradation with different cobalt loading ((**a**), Temperature = 25 °C, pH = 7, PMS concentration = 0.7 mM, catalyst loading = 0.3 g/L, IBU concentration = 10 mg/L); the residual rate of cobalt after adsorbed by each gram of dry resin (**b**).

**Figure 6 materials-15-05003-f006:**
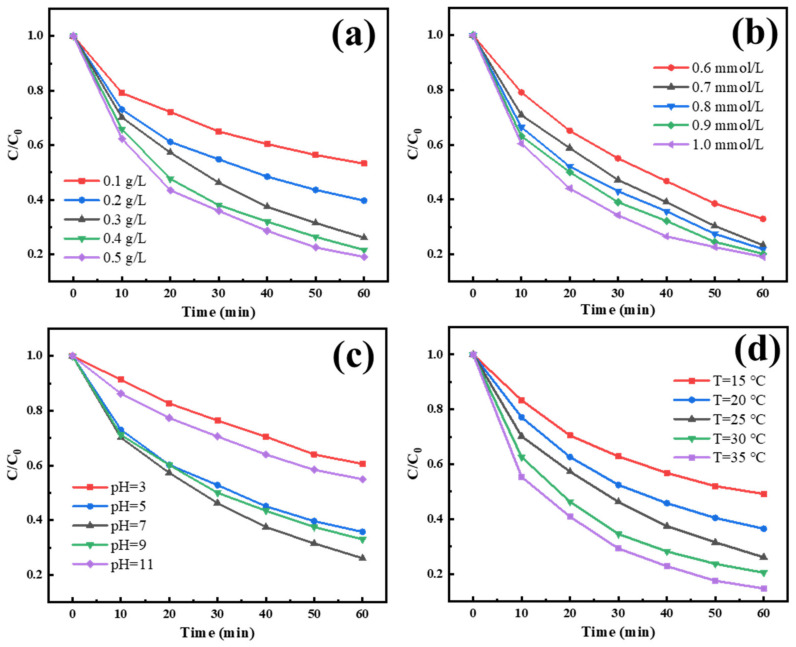
The effects of catalyst loading ((**a**), Temperature = 25 °C, pH = 7, PMS concentration = 0.7 mM, IBU concentration = 10 mg/L), PMS concentration ((**b**), Temperature = 25 °C, pH = 7, catalyst loading = 0.3 g/L, IBU concentration = 10 mg/L), pH value ((**c**), Temperature = 25 °C, PMS concentration = 0.7 mM, catalyst loading = 0.3 g/L, IBU concentration = 10 mg/L) and temperature ((**d**), pH = 7, PMS concentration = 0.7 mM, catalyst loading = 0.3 g/L, IBU concentration = 10 mg/L) on the degradation of ibuprofen, respectively.

**Figure 7 materials-15-05003-f007:**
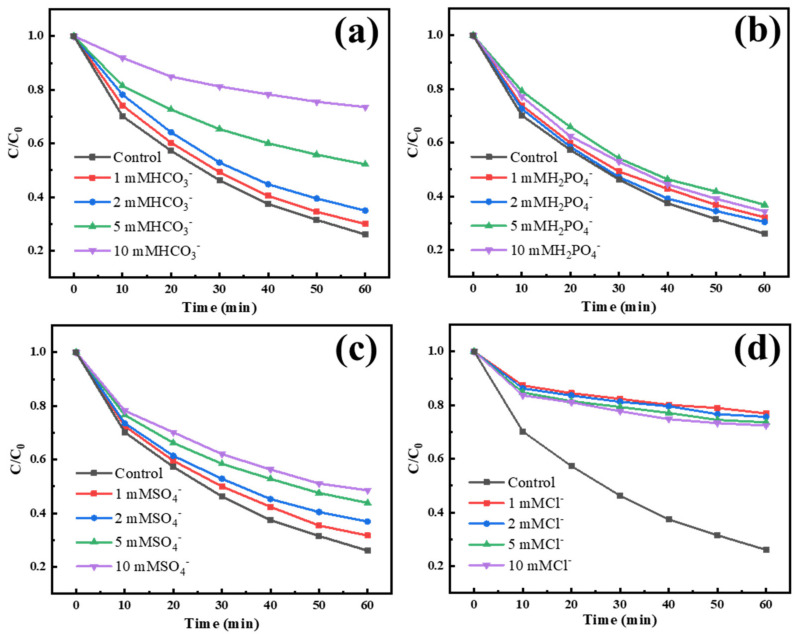
The effects of different concentrations of anions ${\mathrm{HCO}}_{3}{}^{-}$ (**a**), $H_{2}{\mathrm{PO}}_{4}{}^{-}$ (**b**), ${\mathrm{SO}}_{4}{}^{2-}$ (**c**), ${\mathrm{Cl}}^{-}$ (**d**) on the degradation effect of ibuprofen, respectively (Temperature = 25 °C, pH = 7, PMS concentration = 0.7 mM, catalyst loading = 0.3 g/L, IBU concentration = 10 mg/L).

**Figure 8 materials-15-05003-f008:**
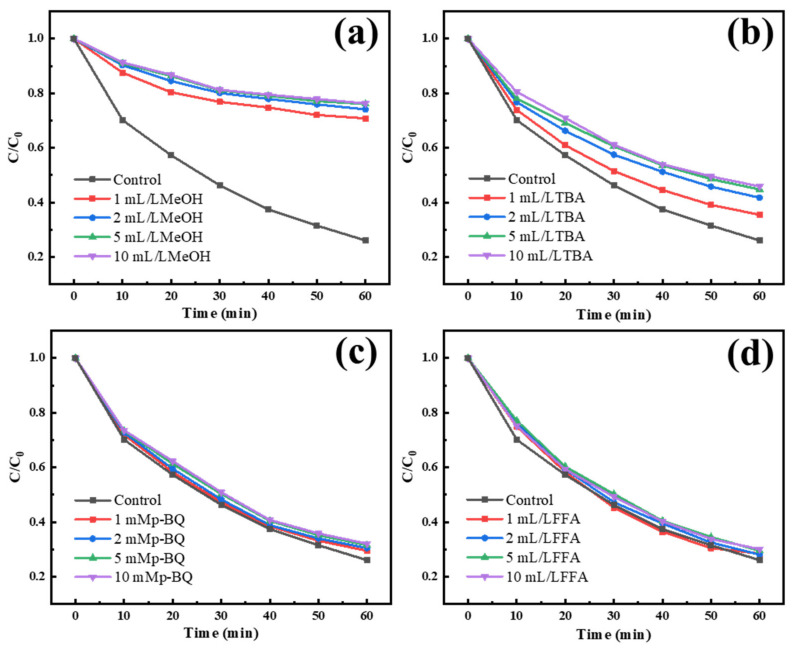
The effect of different quenchers MeOH (**a**), TBA (**b**), p-BQ (**c**), FFA (**d**) on the catalytic degradation of ibuprofen (Temperature = 25 °C, pH = 7, PMS concentration = 0.7 mM, catalyst loading = 0.3 g/L, IBU concentration = 10 mg/L).

**Figure 9 materials-15-05003-f009:**
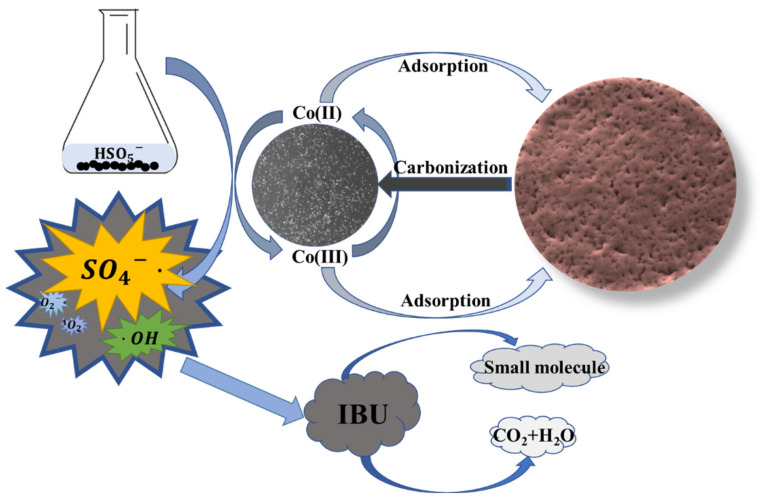
Roadmap for material activated PMS degradation of ibuprofen.

**Table 1 materials-15-05003-t001:** Specific surface area, total pore volume, pore size of several carbon ball materials.

Sample	Surface Area (m^2^/g)	Total Pore Volume (cm^3^/g)	Average Pore Diameter (nm)
New D001 carbon ball	97.6269	0.057726	2.36517
Used D001 carbon ball	67.4460	0.046196	2.73971
Co/Used D001 carbon ball (9 wt%)	31.5425	0.021546	2.73233
Co/Used D001 carbon ball (15 wt%)	19.3434	0.014859	2.78263

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
