# Peer review of "Efficient Activation of Peroxymonosulfate by Cobalt Supported Used Resin Based Carbon Ball Catalyst for the Degradation of Ibuprofen"

_materials, 2022, doi:10.3390/ma15145003_

Round 1
Reviewer 1 Report
The paper reports on the preparation of the catalytic degradation of ibuprofen drug by Co-containing Carbon spheres obtained from resins. While the topic could be interesting, as IBU belongs to the vast category of emerging pollutants, raising environmental concern, the adopted methodology is very basic and the discussion is sometimes simplistic, sometimes unclear. Just to mention some unclear/not scientific sentences: "We speculate that when the used resin is impregnated with cobalt ions, 193 a large number of calcium and magnesium ions are replaced by cobalt ions, and the sur- 194 face is covered by metal composites sintered on it". I think that in a scientific paper, authors should not speculate, but present data and discuss them. It is possibile that Co ions replace other metal ions, ok. There is no quantification of the of the degree of ionic exchange and, then, sintering of "metal composites" is mentioned, but it is not clear the meaning of this sentence. Another example: "Combined with EDS analysis, we found that there are 200 a large number of sulfur and oxygen elements in carbon ball, so the burned metal sub- 201 stances may be the corresponding metal sulfides or oxides": which is the meaning of "large number of sulfur and oxygen elements"... Burning the metal may lead to metal sulfides or oxides? Starting from a -SO3H resins and BURNING it, sulfide form. Then "the metal cobalt can be sintered with 209 the −SO3H in the resin to form metal sulfide." I think that the term "sintering" is not appropriate etc. For these reasons, I think that the paper is not suitable for publication in "Materials".
Reviewer 2 Report
Dear Editor,
The study entitled with "Efficient activation of perxymonosulfate by cobalt supported used resin based carbon ball catalyst for the degradation of ibuprofen" is well designed study and will be usefull for the readers. The qualitiy of the graphs and detailed experiments were presented well. There are some points have to be improved as given below:
-Kinetic calculations have to be added into text to give brief comparasion.
-The results have to be compared with literture and argued.
-The elemenetal content in terms of quantitave results have to be given.
-the pahses of Co have to be clarified.
Round 2
Reviewer 1 Report
The paper has been modified and now can be published.